# Composite Hat Structure Design for Vehicle Safety: Potential Application to B-Pillar and Door Intrusion Beam

**DOI:** 10.3390/ma15031084

**Published:** 2022-01-30

**Authors:** Samer Fakhri Abdulqadir, Faris Tarlochan

**Affiliations:** 1Department of Mechanical Engineering, University of Anbar, Ramadi P.O. Box 5543, Iraq; eq.samer.fakhri@uoanbar.edu.iq or; 2Department of Mechanical and Industrial Engineering, Qatar University, Doha P.O. Box 2713, Qatar

**Keywords:** top-hat section, energy absorption, bending collapse, composite material, door intrusion beam

## Abstract

Regarding crashworthiness, many published works have focused on designing thin-walled structures for frontal collisions compared to side-impact collisions. This paper presents an experimental investigation and finite element modelling of a carbon-reinforced thin-walled top-hat section subjected to quasi-static and dynamic transverse bending loads at different impact speeds. The top-hat sections and their closure assembly plates were made of MTM44 prepreg carbon. The specimens were manufactured by vacuum bagging. Dynamic work was performed to validate the results obtained from the finite element analysis (FEA). The predicted results are in good agreement with the experimental results. The study also showed that the peak load and energy absorption owing to dynamic loading were higher than those under static loading. In the four-point bend analysis, the stacking sequence affected the energy absorption capabilities by 15–30%. In addition, the distance between the indenters in the four-point analysis also affected the energy absorption by 10% for the same impact condition, where a larger distance promoted higher energy absorption. The study also demonstrated that a top-hat shaped thin-walled structure is suitable for deep intrusion beams in vehicle doors for side-impact crashworthiness applications.

## 1. Introduction

The reduction of CO_2_ and other greenhouse gas (GHG) emissions is one of the design constraints of automobile manufacturers [1]. These GHGs are linked to the fuel consumption of vehicles. Heavy vehicles typically consume more fuel and directly contribute to the amount of GHGs released into the environment [1]. Many initiatives have been undertaken globally to reduce GHG emissions from vehicles. Some are focused on developing more engines that are efficient, whereas others are working on reducing the weight of the vehicle. A 10% reduction in vehicle mass can yield a 6–8% reduction in fuel consumption [2]. For vehicle weight reduction, one crucial aspect is ensuring that vehicle safety during collision is not compromised [3,4]. As such, automotive manufacturers are motivated to make their vehicles lighter without compromising vehicle safety by looking at new materials besides the conventional steel used in vehicle chassis and bodies. Therefore, composite materials have received considerable attention.

Many studies have been conducted on the use of composite materials for vehicle crashworthiness, as described in a recent review paper by Isaac et al. [5]. Most of the published work has focused on designing longitudinal tubular vehicle structures for axial and oblique (frontal) crashes without considering side impacts [5,6,7,8,9,10,11]. Carbon fiber-reinforced plastic (CFRP) has been increasingly used over the last decade in many advanced applications owing to its crashworthiness [12,13,14,15,16,17]. Obradovic et al. [18] and Mamalis et al. [19] used the LS-DYNA3D explicit finite element code to model the collapse modes of square carbon fiber-reinforced plastic tubes under static and dynamic tests. Their models were validated experimentally, and good agreement was achieved. The composite material displayed excellent specific energy absorption capabilities during frontal impact. Jiancheng Huang et al. [20] performed numerical and experimental studies to investigate the axial crushing response of a carbon-reinforced composite tube subjected to quasi-static tests. The study concluded that the two-layer finite element model based on the Chang–Chang failure criteria was effective in representing the energy absorption characteristics and crushing failure mode of tubular composite specimens. Jiang et al. [21] described the crush response of a carbon epoxy composite hat shape subjected to axial and oblique crushing loads. The crushing failure of the energy-absorbing elements was predicted using the explicit finite element software ABAQUS through the user-subroutine VUMAT. A complex coupling failure mechanism was revealed, which showed that the cellular structure composites of hat-shaped structures increased the energy absorption capabilities.

Besides frontal collisions, other collisions, such as rear and side impacts, have received less attention compared to frontal collisions. Transverse impacts are a major cause of vehicle accident-related deaths and injuries (Figure 1). The bending deformation of the B-pillars and door intrusion beams and their energy-absorption behavior are major factors that affect the safety of passengers. As reported by the European New Car Assessment Programme (Euro NCAP), side impacts or lateral impacts account for approximately 25% of all crashes. Almost 50% of the occupants injured due to lateral/side impacts are on the opposite side of the one that is struck, giving rise to serious injuries. In view of lowering the weight of the vehicle by using composite materials, it is also important to investigate the crash performance of vehicle-side structures such as B-pillars and intrusion beams made from composites. Several studies have been conducted using metallic structures; however, little work has been conducted on the use of composite materials. The primary failure mode observed was bending deformation of the B-pillars and intrusion beams [22,23].

Hélénon et al. [24] experimentally and numerically investigated the failure of T-shaped laminated composite structures under bending loads. They found that the maximum free-edge principal transverse stresses were at failure locations perpendicular to the fibre direction. Such a design can potentially be used for side effects in vehicles. Kim et al. [25] numerically and experimentally investigated the failure of C-shaped laminated carbon-epoxy laminates. In this study, 20 specimens with different thicknesses and heights were used. They recommended considering the effect of the corner radius in the prediction of curved laminates. Good agreement between the experimental and predicted results was obtained. Liu et al. [26] presented an experimental investigation of the failure modes and effect of thickness on the crashworthiness of a double-hat tube made of weave carbon fiber-reinforced plastic under quasi-static axial loading and transverse bending tests. They concluded that the thickness increased the peak load and energy absorption in both axial and transverse tests. The double-hat shaped tubes (irregular shape) had higher energy than most regular sectional CFRP tubes and twice that of conventional metallic tubes. The study showed that the specific energy absorption and a peak load of the transverse bending were less than 1% and 10% than those of the axial loading.

Koricho et al. [27] numerically and experimentally investigated the behavior of a woven fabric carbon/epoxy composite T-joint. The study showed that the predicted results for the load-carrying capacity and stress distribution of the T-joint were in good agreement with the experimental results. The study concluded that numerical simulation is an effective tool for studying the behavior and performance of automotive structural joints without constructing expensive prototypes. Pavan et al. [28] numerically analyzed three different sections, circular, M-shaped, and hat sections, for various combinations of shapes and thicknesses. It was found that the hat-shaped cross-section is optimum for the door beam, which gives better energy absorption with comparatively less weight; however, the study was limited to metallic structures. In view of the literature review, it was found that almost no research has reported on top-hat structures made from polymer composites subjected to transverse loading. Hence, the purpose of this study was to further the work of Pavan et al. [28], who investigated thin-walled top-hat made from carbon composites to promote a lightweight structure. Most of the reported work on the transverse loading of such thin-walled structures focused on quasi-static loading experiments. This work presents new findings related to dynamic loading. The paper is organized as follows: (a) materials and methods: describing the sample preparation and experimental setup; (b) numerical modelling, describing the finite element model and validation process; and (c) results and discussions followed by a conclusion.

## 2. Materials and Methods

This section describes the fabrication of composite top-hat structures, along with the transverse loading (bending) experimental setup for both dynamic and static loads. The section begins by explaining the materials used in this study, followed by the geometric design of the samples. The manufacturing process is explained, followed by the experimental setup.

### 2.1. Materials

Prepreg carbon-reinforced composite materials are becoming increasingly common in the composite industry owing to their ease of use, consistent properties, and high-quality surface finish. The “prepreg” term is an abbreviation for the phrase pre-impregnated. A prepreg is a reinforcing fabric pre-impregnated with resin. The prepreg is prepared to be laid and stuck into the mould without the addition of any resin, and epoxy resin is most often used in industries. Prepreg carbon fibre fabric is a traditional carbon fibre fabric that has been pre-impregnated with uncured resin during manufacture. Because the resin has already been mixed with its hardener, the prepreg carbon fibre must be stored at a very low temperature range to prevent the resin from curing before the prepreg is used. Prepreg fabric is normally stored at approximately −20 °C. The mechanical properties of the prepreg depend on the ratio between the resin and the reinforcement, and the properties change as the ratio changes. For this study, carbon fibre prepreg and epoxy MTM44/CF5804A−40% RW-DC were utilized. This material has a high performance at 180 °C curing, and it was designed to be manufactured using autoclave molding or a low-pressure vacuum. The volume fraction of the fibre was 60% with a mass of 285 g/m^2^, the resin content was 40%, and the overall prepreg mass was 475 g/m^2^.

### 2.2. Geometry of Top Hap Structure

The tool used as a mould was made of aluminum 5754 series with a thickness of 2.5 mm and a length of 800 mm. The flange width was set at 75 mm on both sides. Nine aluminium ribs made of the same material and thickness were used. The ribs were placed and welded beneath the tool to support it and to avoid any bending that might occur when sticking the prepreg composite in the tool. An extra support was obtained by attaching two metal rods, which were welded to the ribs. The purpose of these rods is to hold the ribs by preventing any movement when laying the composite. The distance between the ribs was set to 100 mm, as shown in Figure 2. Two different radii were used in the specimen geometry: the first radius was 8 mm, which represented the upper radius near the flange, while the second one was 11 mm, which represented the lower radius near the top wall of the specimen. The flange length of the specimens was set to 26 mm. The top wall was set at 63 mm, and the sidewall was set at 40 mm, excluding the radii dimension. The composite material was cut, and the specimen was manufactured with a length of 550 mm and a width of 270 mm.

### 2.3. Manufacturing of Top-Hat Structure

Figure 3 depicts the fabrication process of the top-hat-shaped tubes using vacuum bagging, which consists of prepreg carbon placed in a metal mould (Figure 2). The mould (tool) was designed with a large flange area, which produces plenty of space to stick the plies firmly, as well as a stick release film, breathing layer, and bagging tape. A polytetrafluoroethylene (PTFE) release film layer was laid on the tool to avoid sticking the part with the tool during the process. The top-hat section and closure assembly plate were manually laid ply by ply. The top-hat and closure plates were fabricated separately. The ply orientations for both the top-hat and closures were stacked in sequence using eight plies of woven laminates as an isotropic layup of (45/−45/90/0/0/90−45/45)_2_ stacking sequence. The 0-degree layer had the same fibre direction as the longitudinal tool direction. The 45° orientation was placed at the top and bottom surface ply of both the part and the closure plate.

A vacuum bagging cure was used to manufacture the top-hat and the closure plate. Wrinkles and folding of materials in highly contoured parts were observed, especially in concave areas. The manufacturer recommends two cycles of curing for 2 h at 130 °C, followed by 2 h at 180 °C with a maximum cooling rate of 3 °C/min. A Sika-Power 490C adhesive was used to join the parts with the closure plates. The adhesive has an embedded glass ball (0.3 mm to maintain a uniform thickness along the joined area. The contact area of both the part and closure plate was roughened with an abrasive material before applying the adhesive to increase the friction and remove any oily area which might affect the joining area, as instructed. Grips were used to remove bubbles and maintain consistent thickness and distribution.

After joining the top-hat and closure assembly plate, the specimen was left for four hours at ambient temperature, after which the specimen was placed in an oven at 85 °C for 30 min for curing. The top-hat part length was manufactured as 550 mm, which was shortened to 450 mm for the three-point bend test. Both flanges on the side of the top-hat were shortened to 25 mm and the closure assembly was trimmed to fit the top-hat length. A total of 12 top-hat structure samples were prepared for three different dynamic and quasi-static loadings.

### 2.4. Testing

#### 2.4.1. Coupon Testing—Material Characterization

Carbon fiber-reinforced composite MTM44 was used in this study. Composite plaques with dimensions of 500 × 500 mm were processed via oven vacuum bagging to produce the samples. Six plies, with an orientation of ±45°, were manufactured for each plaque. The fibre weight (volume fraction) was 0.6 and the prepreg weight was 475 g/m^2^. The plaques were cured at 135 °C for two hours and followed by two hours at 180 °C. After curing, the plaques were slowly cooled at a rate of 3 °C/min. The samples were cut using a composite precision cutting machine. Two sets of samples were cut according to the stated orientation. The first set was cut with an orientation of (0/90)° and the second set was cut to produce a fibre orientation of ±45°. A uniaxial tension test was performed in accordance with the test method ASTM D30309/D3039M [29]. Tabs were applied on both ends of the (0/90)° samples because this orientation withstood a higher load than the ±45° orientation did. Instron 5800R 100 kN Test machine was used to test the samples with a load cell of 100 kN. Two transducers were positioned in the middle of the samples to measure the longitudinal and transverse strain, as shown in Figure 4. The test was performed at room temperature with a quasi-static velocity of 2 mm/min. The material properties are listed in Table 1 and were used to define the finite element model.

#### 2.4.2. Three-Point Bending Test (TPBT)

A three-point bending test was used for the experimental phase owing to the availability of test fixtures in the laboratory. The main objective of TPBT is to validate the finite element model. Two different types of equipment were used in this study depending on the type of loading. The dynamic test was implemented using an Instron spring-assisted instrumented drop tower. The instrument was designed for different impact crushing energies of up to 10 kJ. The total impact mass used was 97 kg (drop weight, bolt weight, and drop-tower carriage weight). A bending fixture was placed and fixed with screws in the drop tower at the lower platen. The span width was 300 mm [22,23,26], and the crosshead diameter was 80 mm. A lower support diameter corresponds to where the specimen was placed at the centre of the bending fixture. The crosshead impactor was fixed upward at the moving platen of the drop-tower fixture. The length of the specimen was set to 450 mm. The tests were conducted at room temperature. A high-speed camera (FASTCAM SA-X2) with a rate of 500 frames was used to capture and record the crushing of the specimens during the test. The following impact speeds were used (2.13, 3.0 and 3.5 m/s). The quasi-static test was carried out using Instron 5800R 100 kN equipment.

The objective of conducting the quasi-static test was to compare the results with those of the dynamic test and determine the strain rate sensitivity. The specimens were subjected to a quasi-static load at a rate of 2 mm/min at room temperature. A load cell of 100 kN was used. The force-displacement curves were recorded using computer software, while the energy-displacement curves were calculated according to the force-displacement data.

## 3. Numerical Modelling

The objective of developing a finite element model in this study was to investigate the four-point bending behaviour and energy absorption performance of the top-hat structure at different indenter speeds, distances between indenters, and laminate stacking sequences. The four-point bending test was selected in this study because it is in agreement with Saint-Venant’s theorem [30] compared with the three-point bend test. Because the available experimental equipment only permits three-point bending, the experiment was conducted using this setup to validate the finite element model. Subsequently, a finite element model was used to simulate the four-point bending in a dynamic scenario (Figure 5). The commercial nonlinear finite element code ABAQUS was employed in this study to predict the behaviour of a carbon-fiber-reinforced closed top-hat structure. In this study, the top-hat structure is assumed to have a thin-walled shell structure. Therefore, the top-hat profile and closure assembly plate were modelled using the linear shell element S4R node doubly curved, hourglass control, and reduced integration with finite membrane strains. This shell element, S4R, has both bending and membrane capabilities that permit in-plane and normal loads. Reduced integration provides a significant reduction in the computational time without a loss of accuracy [27]. Based on the mesh sensitivity reported in previous studies [31], an element size used was 5 mm.

To model the composite material in ABAQUS, the property category of the layup editor provides options that allow the user to define every ply in the laminate separately, including the ply thickness, material, name, and orientation. The Hashin damage criterion was used to investigate the failure [27,31]. In this failure criterion, the following failure modes are considered: (a) mode 1, fibre breakage in tension; (b) mode 2, fibre buckling; (c) mode 3, matrix cracking; and (d) mode 4, matrix crushing [27]. The failure initiation criteria for the aforementioned failure modes are as follows:(1)$$F_{f}^{t}={\left(\frac{\sigma_{11}}{X^{T}}\right)}^{2}+\alpha {\left(\frac{\tau_{12}}{S^{L}}\right)}^{2}$$

For Mode 1.
(2)$$F_{f}^{c}={\left(\frac{\sigma_{11}}{X^{C}}\right)}^{2}$$

For Mode 2.
(3)$$F_{m}^{t}={\left(\frac{\sigma_{22}}{Y^{T}}\right)}^{2}+\alpha {\left(\frac{\tau_{12}}{S^{L}}\right)}^{2}$$

For Mode 3.
(4)$$F_{m}^{c}={\left(\frac{\sigma_{22}}{2S^{T}}\right)}^{2}+\left[{\left(\frac{Y^{C}}{2S^{T}}\right)}^{2}-1\right]\left(\frac{\sigma_{22}}{Y^{C}}\right)+{\left(\frac{\tau_{12}}{S^{L}}\right)}^{2}$$

For Mode 4, where *X^T^* and *X^C^* are the longitudinal strengths (tension and compression respectively), *Y^T^* and *Y^C^* are the transverse strengths (tension and compression, respectively), *S^L^* and *S^T^* are the shear strengths in principal material directions. *α* is a coefficient that represents the effect of shear stress on fibre tensile failure initiation. *σ*_11_, *σ*_22_, *τ*_12_ are stress tensors.

The composite laminate was modelled using a lamina variable in addition to other variables, which are described in terms of composite parameters, including longitudinal tensile and compression strength, transverse tensile and compression, and longitudinal tensile and compression shear (Table 1). As shown in Figure 5, the initial finite element model for three-point bending was developed for validation with the experimental data. The indenter and two supports were modelled as a linear discrete rigid element R3D4 bilinear quadrilateral. The profile length was 450 mm and the distance (span) between the two supports was 300 mm. The number of elements for the entire structure was approximately 7000 shell elements, and the degrees of freedom were six. The boundary conditions at the two supports were fixed in all directions and the motion was constrained in all degrees of freedom (U1 = U2 = U3 = UR1 = UR2 = UR3 = 0). The indenter (load) was restricted to move vertically downward toward the profile, while other directions were fixed (U1 = U3 = UR1 = UR2 = UR3 = 0), as shown in Figure 5.

A general contact interaction was set for the entire model. This type of contact is used to avoid interpenetration of the structure wall. A finite sliding penalty with contact pairs and ‘hard’ contacts was used to define the contact of the entire structure. Based on previous studies conducted by the authors, the friction coefficient value was set as 0.2 of all contact surfaces [3,31]. The closure plate was assembled with a top-hat profile by applying an adhesive layer. This was achieved by using a tie constraint. A tie constraint ties two separate surfaces together so that there is no relative motion between them. Tie constraint allows you to fuse together two surfaces even though the meshes created on the surfaces may be dissimilar. The plate was selected as the master surface. A node-based surface was created for this interaction. The diameter of the indenter, which represents the load applied to the profile, was set to 80 mm, and the length was set to be along the width of the structure, while the two support diameters were set at 40 mm with semi-circle geometry to decrease the mesh as well as the time required for each iteration. The support length was set along the profile width. The applied velocity was set to be quasi-static at a rate of 2 mm/min, whereas in dynamic loading, the velocity was set according to the required speed and assigned with an impacting mass at the indenter. The mass was set at 97 kg for all dynamic test speeds. The orientation of the fibers for both the top-hat and closure plates was set as (45/−45/90/0/0/90−45/45)_2_ which matches the experimental arrangement. Sixteen plies were used to simulate the composite material, with three integration points. The total thickness of the composite was approximately 2.6 mm for both the top-hat section and the closure plate assembly. For the four-point bend test, a similar FE model developed for the three-point bend test was utilized with the addition of an indenter, as shown in Figure 5. The parameters studied in the four-point bend test are listed in Table 2.

## 4. Results and Discussion

The results section presents the validation results of the FE model. This validation was performed for both static and dynamic loadings. This is based on a three-point bending simulation. As discussed in Section 2 of this paper, these FE models were used to analyze four-point bending considering the speed of the indenter, the distance between the indenters, and the stacking sequence of the laminate used. The last part of this Section 4 will discuss the findings obtained from the four-point bend test (simulation).

### 4.1. Validation of FE Model

#### 4.1.1. Dynamic Loading Validation

The load-displacement and energy absorption results for the carbon fiber reinforced closed top-hat specimen subjected to three different dynamic speeds (2.13, 3.0 and 3.5 m/s) are shown in Figure 6. These three dynamic speeds correspond to theoretical impact energies of 220, 440, and 600 J, respectively. The specimens were subjected to a dynamic speed with an impact mass of 97 kg and a span of 300 mm, and the impact roller diameter was 80 mm. The peak load increased with the impact speed. The repetition of cracks in various locations led to a decrease in the load as the test progressed. This caused fluctuations, which corresponded to crack propagation along the impactor axis.

The crack began at the top wall corners, which was caused by stress concentration due to the impactor, as shown in Figure 7. The region beneath the impactor suffered two opposite loadings, because the top wall was subjected to compression loading and the bottom wall suffered from tension. The crack propagated along the impactor axis. Owing to the increase in compression loading, dislocations of the fibers and matrix appeared in the cracking area. The crack spread along the top wall and sidewall beneath the impactor, which led to buckling of the sidewall of the structure (Figure 8).

The finite element predictions for the load-displacement curves are displayed in Figure 9, Figure 10 and Figure 11. The predicted peak loads for all speeds were lower than the experimental values; subsequently, the predicted load became closer to the experimental value. The predicted load from the numerical analysis is similar to the experimental results. For the predicted energy-displacement curve for the dynamic speeds, the behavior of the predicted energy corresponded with those of the experimental tests. The energy increased almost linearly and rebounded at the end of the test owing to the non-deformed closure plate.

Figure 12 shows a comparison between the experimental and predicted results of the composite closed-top hat subjected to different transverse dynamic speeds. The figure shows good agreement between the experimental and predicted results from the FE model, and the behaviors of both appear to be consistent. The predicted results, experimental results, and the differences between them are listed in Table 3.

#### 4.1.2. Quasi-Static Validation

The quasi-static load-displacement curve of the carbon fiber-reinforced closed top-hat section is presented in Figure 13. The specimens were subjected to bending loading at a constant speed of 2 mm/min. The span was 300 mm and the crosshead diameter was 50 mm. The top wall of the structure was subjected to compression and bottom wall tension, while both sidewalls suffered from combined tension and compression load, and the stress was concentrated at the corner of the top wall. From the load-displacement curve, it can be noted that the load increased steadily, and the specimen behaved elastically until it reached the highest value (point A). A drop-in load can be observed until it reaches point B, owing to a crack in the upper corner of the top wall and failure of the matrix. An additional drop in load can be seen at point C, which is attributed to a crack in the sidewall of the structure underneath the impactor that started and propagated until total failure of the structure occurred. The energy was absorbed owing to the deformation of the specimen, and the energy was calculated by measuring the area under the load-displacement curve. The specimens under a static load were able to absorb approximately 230 J with a deformation length of 13–16 mm.

### 4.2. Finite Element Analysis Four-Point Bending

Figure 14 and Figure 15 depict the load-displacement diagrams based on the design configurations stipulated in Table 2. From Figure 16, for the same speed and stack sequence, the peak force in (L2) is higher when the distance between the two indenters is 150 mm, which is attributed to the centers of the indenter near the rigid support cylinder, and the reaction will be higher. Energy absorption was also higher because the mean force was higher. For the same distance between the indenters, a higher speed influences the energy absorption and mean force, and it increases as the speed increases. For the same distance and speed, the energy absorption and mean force are higher at 45/−45 than at 0/90, and it increases as the speed increases. This is attributed to the structure with 45/−45 stuck, revealing a little stiffness compared to other stack sequences. In general, 45/−45 (L2) shows higher energy absorption and mean force than the 0/90 (L1) stack sequence.

## 5. Conclusions

The study exhibited an experimental investigation and finite element predictions of the dynamic and quasi-static behavior of carbon fiber-reinforced top-hat-shaped tubes. The load-displacement and energy-displacement curves for both dynamic and quasi-static loads, as well as the different dynamic impact speeds, are discussed. The differences between the dynamic and quasi-static peak loads and energy absorptions were compared. The conclusions are summarized as follows.

The energy absorption in the dynamic test increased linearly with an increase in the impact speed. The primary mode of energy absorption is the bending of the structure and failure of the composite material through micro fragmentation and splaying.Energy absorption due to dynamic loading was higher than those with static loading at the same deformation length.The predicted results are in good agreement with the experimental tests. The Hashin material model used in finite element analysis is applicable for evaluating composite structures under dynamic transverse loading scenarios.Under dynamic loading, the top-hat structures displayed good energy absorption capabilities. This is a crucial factor when designing structures for crashworthiness applications.Four-point bending analysis showed that the distance between indenters does affect the crash performance.

The implications of the present result shows the promising usage of top-hat structures for energy absorption applications. Such structures are simple to manufacture and are lightweight, with promising crashworthiness capabilities. The future steps of this work could be: (i) looking at different failure models besides Hashin such as composite degradation model and progressive failure analysis; (ii) optimizing the design, and (iii) developing B-pillar and door intrusion beam design guidelines after finite element analysis of a car door and tested as per side-impact test regulations such as FMVSS 214.

## Figures and Tables

**Figure 1 materials-15-01084-f001:**
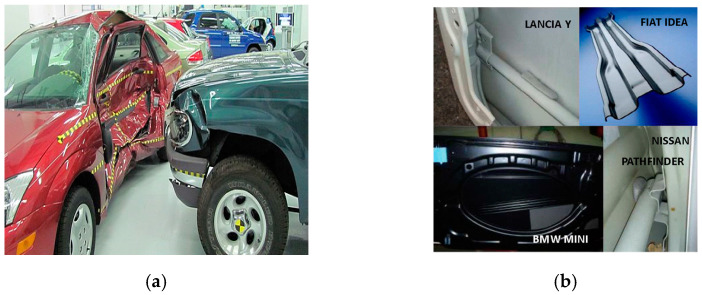
(**a**) Example of side impact between vehicles (source of image: Brady Holt, CC BY 3.0, via Wikimedia Commons). (**b**) Example of door intrusion beams/side impact beams (source of image: Strano. m, CC BY 3.0, via Wikimedia Commons).

**Figure 2 materials-15-01084-f002:**
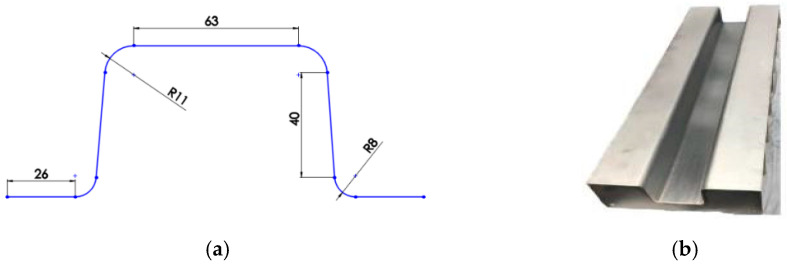
(**a**) Tool geometry as a mold for preparing composite samples (**b**) fabricated mold.

**Figure 3 materials-15-01084-f003:**
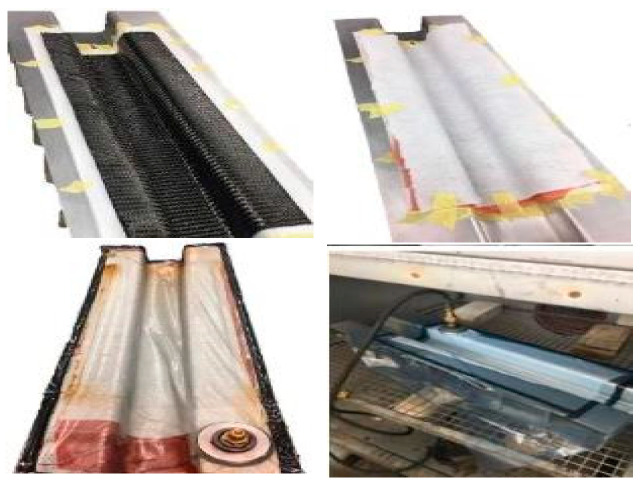
Composite manufacturing procedures. The top figures display the layup procedure, while the bottom figures display the vacuum bagging preparations.

**Figure 4 materials-15-01084-f004:**
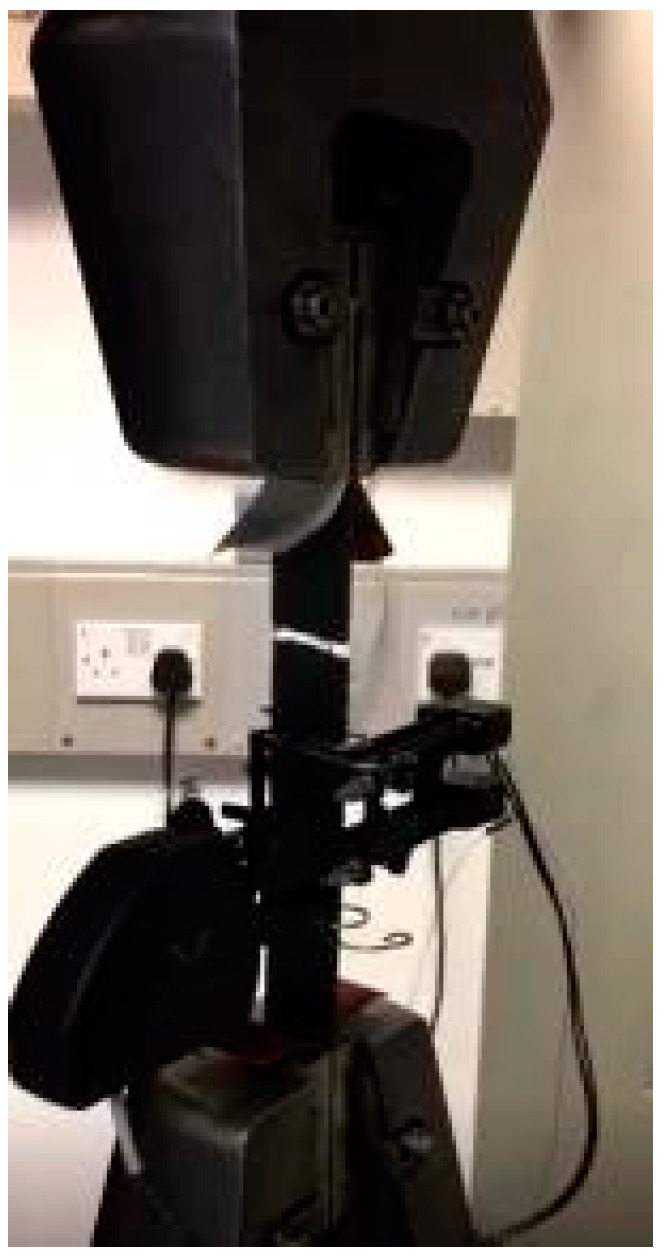
Composite samples under quasi-static loading with strain transducers.

**Figure 5 materials-15-01084-f005:**
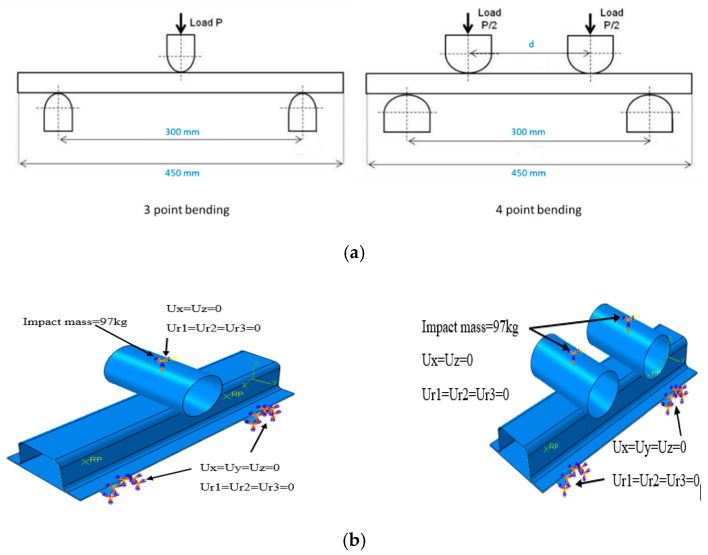
(**a**) Three-point and four-point bending setup used in the study. (**b**) Finite element (FE) model of top-hat geometry with boundary conditions.

**Figure 6 materials-15-01084-f006:**
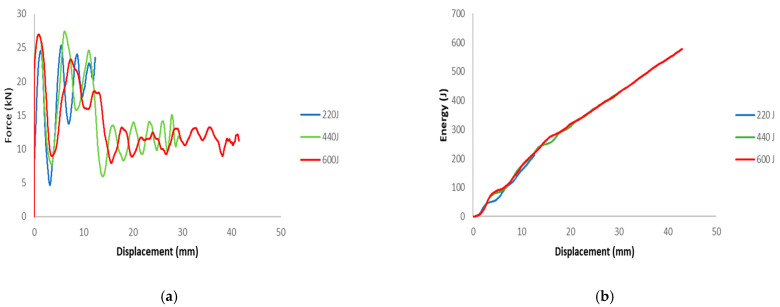
Representative results of (**a**) force-displacement and (**b**) energy-displacement for dynamic experimental test.

**Figure 7 materials-15-01084-f007:**
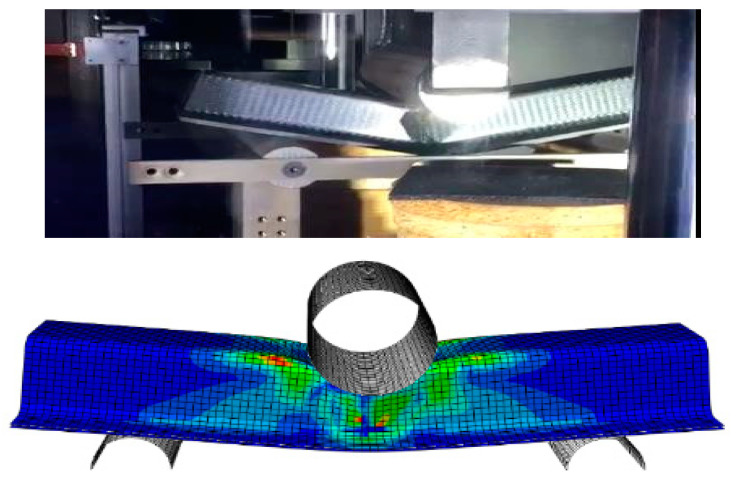
Experimental and Finite element dynamic loading.

**Figure 8 materials-15-01084-f008:**
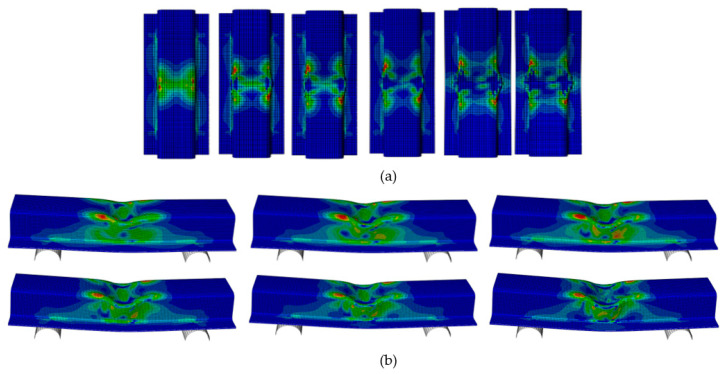
Different time steps for dynamic transverse loading at a speed of 3.5 m/s, (**a**) top view (**b**) side view.

**Figure 9 materials-15-01084-f009:**
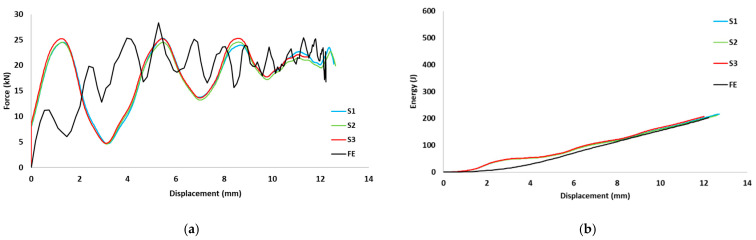
(**a**) Experimental and predicted load vs. displacement curves, (**b**) Energy vs. displacement for hat-shaped crush at crush speed of 2.13 m/s.

**Figure 10 materials-15-01084-f010:**
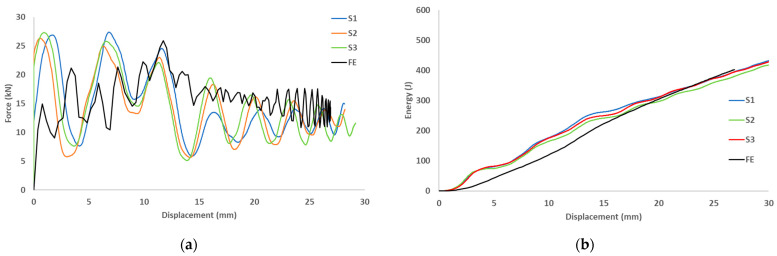
(**a**) Experimental and predicted load vs. displacement curves, (**b**) Energy vs. displacement for hat-shaped crush at crush speed of 3 m/s.

**Figure 11 materials-15-01084-f011:**
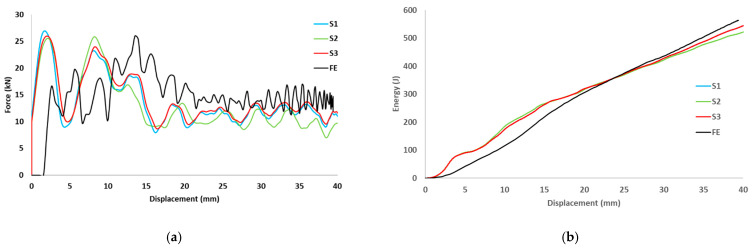
(**a**) Experimental and predicted load vs. displacement curves, (**b**) Energy vs. displacement for hat-shaped crush at crush speed of 3.5 m/s.

**Figure 12 materials-15-01084-f012:**
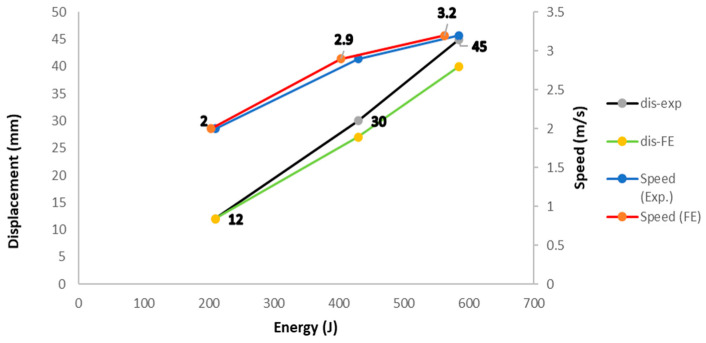
Comparison of the energy-speed curves between experimental test and numerical model.

**Figure 13 materials-15-01084-f013:**
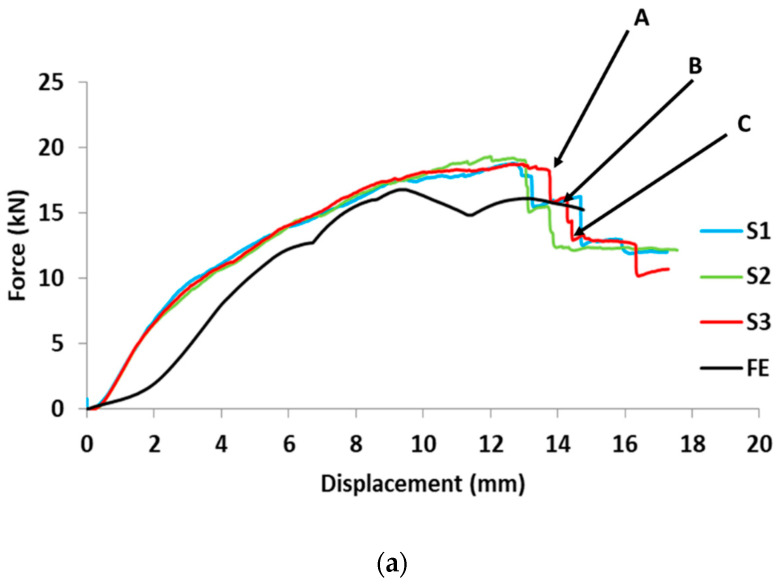
(**a**) Load-displacement diagram for static experimental and finite element model and (**b**) Finite element and experimental quasi-static crushing of structure.

**Figure 14 materials-15-01084-f014:**
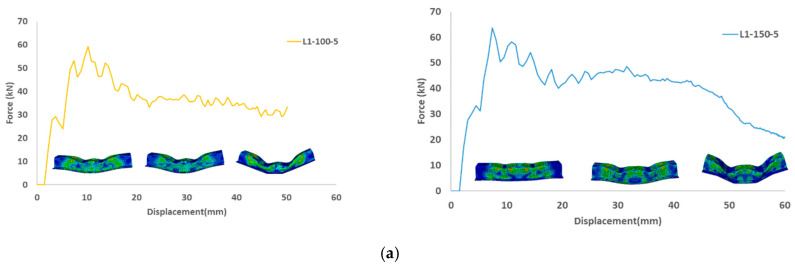
Load displacement diagrams at crushing speed of 5 m/s for different indenter distance of 100 mm and 150 mm (**a**) laminate configuration ${\left(0/90/0/90/0/90/0/90\right)}_{2}$ (**b**) laminate configuration ${\left(45/-45/90/0/0/90-45/45\right)}_{2}$.

**Figure 15 materials-15-01084-f015:**
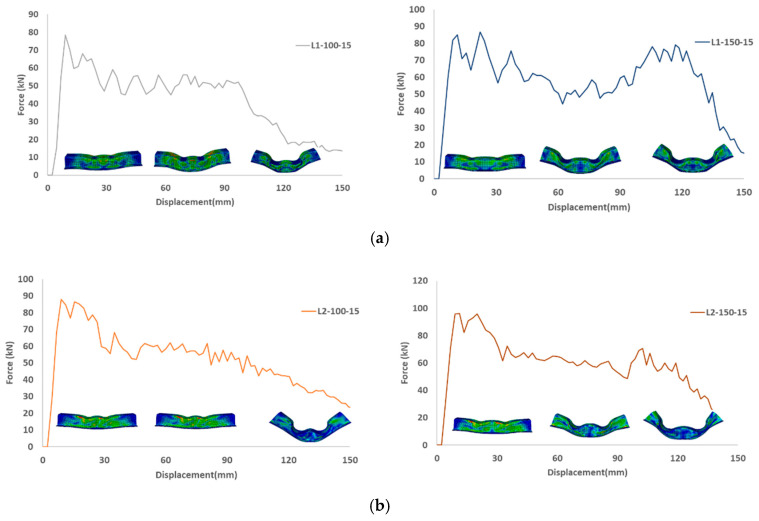
Load displacement diagrams at crushing speed of 15 m/s for different indenter distance of 100 mm and 150 mm (**a**) laminate configuration ${\left(0/90/0/90/0/90/0/90\right)}_{2}$ (**b**) laminate configuration ${\left(45/-45/90/0/0/90-45/45\right)}_{2}$.

**Figure 16 materials-15-01084-f016:**
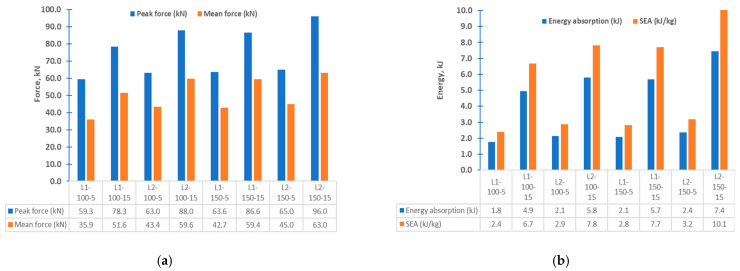
Summary of findings for four-point bend crushing (**a**) peak and mean force, (**b**) energy and specific energy absorption.

**Table 1 materials-15-01084-t001:** Mechanical properties of MTM44/CF5804A-40%RW-DC.

Material Property	Symbol and Units	Value
Young’s modulus, tension in direction	$E_{11}$(GPa)	70
Young’s modulus, compression in direction	$E_{22}$(GPa)	70
Shear modulus in plane 1–2	$G_{12}$(GPa)	4
Poisson’s ratio in plane 1–2	μ_12_	0.05
Tensile strength in direction 1/(MPa)	X^T^ (MPa)	760
Compressive strength in direction 1	X^C^ (MPa)	608
Shear strength in plane 1–2	S^T^ = S^L^ (MPa)	100

**Table 2 materials-15-01084-t002:** Parameters studies through four-point bend test.

Parameters	Level 1 (L1)	Level 2 (L2)
Stacking sequence	${\left(0/90/0/90/0/90/0/90\right)}_{2}$	${\left(45/-45/90/0/0/90-45/45\right)}_{2}$
Distance between indenter (d)	100 mm	150 mm
Indenter speed	5 m/s	15 m/s

**Table 3 materials-15-01084-t003:** Predicted and experimental results for composite material reinforced closed top hat.

Speed (m/s)	Theoretical Energy at Impact (J)	Experimental (J)	Numerical (J)	Error %
2.13	220	215	203	5.5
3.0	440	435	413	5.0
3.5	600	580	564	2.7

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
