# Peer review of "Composite Hat Structure Design for Vehicle Safety: Potential Application to B-Pillar and Door Intrusion Beam"

_materials, 2022, doi:10.3390/ma15031084_

Round 1
Reviewer 1 Report
The manuscript “Composite Hat Structure Design for Vehicle Safety: Potential Application to B-Pillar and Door Intrusion Beam” is well written and logically structured. Both numerical and experimental results are provided. The reviewer recommends the publication after minor revision. My comments are listed as below:
More details should be added into Figure 3. For example, subfigures demonstrating manufacturing process.
Figure 4 is not very clear (too dark).
Figure 5 (a) Three-point and Four-point instead of 3 point and 4 point.
In terms of Figures 6, 9, 10, 11, 12, 14, and 15, please put legends into the figures.
In Conclusion, it is better to mention the future work explaining the work that cannot be done at the moment.
Author Response
The manuscript “Composite Hat Structure Design for Vehicle Safety: Potential Application to B-Pillar and Door Intrusion Beam” is well written and logically structured. Both numerical and experimental results are provided. The reviewer recommends the publication after minor revision. My comments are listed as below:
More details should be added into Figure 3. For example, subfigures demonstrating manufacturing process.
Reply from Authors: Thank you for suggestion. We have added in the caption to explain further on the manufacturing process.
Figure 4 is not very clear (too dark).
Reply from Authors: Thank you for suggestion. Unfortunately, this is the only figure we have. However, we will wait for the comments from the editorial team. If required, we will remove the figure.
Figure 5 (a) Three-point and Four-point instead of 3 point and 4 point.
Reply from Authors: Thank you for suggestion. The correction was made in the manuscript.
In terms of Figures 6, 9, 10, 11, 12, 14, and 15, please put legends into the figures.
Reply from Authors: Thank you for suggestion. Legends are provided in the figures.
In Conclusion, it is better to mention the future work explaining the work that cannot be done at the moment.
Reply from Authors: Thank you for suggestion. A small write-up was added to the conclusion section.
Reviewer 2 Report
1. This work mainly describes FE-analysis of composite structure under dynamic load. But a lot of paper related to this topic are published in recent year, where progressive damage analysis, and 2D and 3D Hashin model are used for FE-analysis of composite structure under dynamic load. So you should explain clearly your novelty.
2. This work is related to potential application of automotive part. But the manufacturing process considered in this study, like vacuum bag, can be used for mass production? This process needs long curing time as you mentioned in section 2.3.
3. Why did you consider both 3 and 4 point bending? These bending tests show similar deformation behavior.
4. How did you obtain Yt, Yc, Sl, Sc and alpha in Hashin failure criterion in Section 3?
5. How did you deal with adhesive layer in FE-model between top-hat section and closure plates?
6. How did you deal with crack propagation in FE-model?
Author Response
- This work mainly describes FE-analysis of composite structure under dynamic load. But a lot of paper related to this topic are published in recent year, where progressive damage analysis, and 2D and 3D Hashin model are used for FE-analysis of composite structure under dynamic load. So you should explain clearly your novelty.
Reply from Authors: Thank you for the comment. The authors agree with the reviewer regarding the extensive work done using FE-analysis. However, the focus of the paper is the extension of the work done by Pavan et. al. where he and he co-researchers had investigated using conventional metal top-hat structures and found promising results. As such, in this current study the authors are continuing the work of Pavan by investigating thin-walled top-hat structures made from carbon composites. We are also extending the work of Pavan from door intrusion beams to add on the B-pillars as well. Therefore, this study is just an initial study of top-hat composite structures for transverse loading applications.
Pavan K, S. Nagarajan, Enhancement of car door optimization for crash analysis, Int. J. Pure Appl. Math. 2018, 119–7, 1039–1045
- This work is related to potential application of automotive part. But the manufacturing process considered in this study, like vacuum bag, can be used for mass production? This process needs long curing time as you mentioned in section 2.3.
Reply from Authors: Thank you for comment. This is a valid point. Mass production has always been a barrier to the usage of composite lightweight material in the transportation industry. Due to the limited availability of manufacturing equipment in the lab, vacuum bagging was selected to produce composite samples of high quality without bubble entrapment between layers (and other manufacturing defects found from hand layups). The goal is not to access the manufacturing limitation but rather the performance of such structures under transverse loading. There are many entities researching into mass production of composite such as The Aachen Center for Integrative Lightweight Production (Achener Zentrum für integrativen Leichtbau, or AZL) of RWTH Aachen University (Aachen, Germany) and Cannon. Lately Cannon has produced a technology for mass production of composite parts (Reinforced Plastics, Volume 60, Number 5, September/October 2016)
- Why did you consider both 3 and 4 point bending? These bending tests show similar deformation behavior.
Reply from Authors: Thank you for suggestion. The four point bending test was selected in this study because it is in agreement with the Saint-Venant’s theorem [Paulo et. al], compared with the three-point bend test. Besides this, due to the distance between the indenters in four-point bend test the deformation is more compared to three point, giving an higher energy absorption characteristics compared to the three point bend test. The results from this work suggests that designing such structures as three point bend loading underestimates the actual capabilities of such structures. This is an important element to be considered in designing such beams to be included in the vehicle door intrusion beams and possibility as B-pillars.
Paulo Ricardo Gherardi Hein, Loïc Brancheriau. Comparison Between Three-Point And Four-Point Flexural Tests To Determine Wood Strength Of Eucalyptus Specimens. Maderas. Ciencia y tecnología, 2018, 20(3): 333 - 342. DOI: 10.4067/S0718-221X2018005003401
- How did you obtain Yt, Yc, Sl, Sc and alpha in Hashin failure criterion in Section 3?
Reply from Authors: Thank you for the question. Yt: was calculated by taking the average (Yt) of the samples in (0/90) used in the study as follows = (MAX (P1: Pa))/((W*t)) where a is the iteration of the applied load, W is the width of the sample and t is the thickness of the sample. For SI: Yt: was calculated by taking the average (Yt) of the samples in (± 45) used in the study as follows = (MAX (P1: Pa))/(2*(W*t)). Finally, α: was calculated by dividing the minor strain (transverse direction) to the major strain in the force direction within elastic region.
- How did you deal with adhesive layer in FE-model between top-hat section and closure plates?
Reply from Authors: Thank you for suggestion. It was modelled as cohesive interaction.
- How did you deal with crack propagation in FE-model?
Reply from Authors: Thank you for question. In the Hashin model used in ABAQUS, crack propagation was not modelled. Here in Hashin model, damage initiation refers to the onset of degradation at a material point. We use the Hashin criteria to evaluate the propensity of the material to undergo damage without modeling the damage process.
Reviewer 3 Report
The authors present the article entitled “Composite Hat Structure Design for Vehicle Safety: Potential Application to B-Pillar and Door Intrusion Beam”. However, the manuscript in this current form presents the next concerns:
The manuscript is found with some grammatical and typographical errors. The authors are suggested to go through the manuscript thoroughly and proofread for grammatical and typographical errors.
Lines 23-28: References are missing.
The objective of the manuscript is ambiguous. Please re-write it for better comprehension. Also, I suggest writing the objective according to the novelty of the manuscript. Is it a research or state-of the art review paper?
The Introduction section is not impressive. A detailed study is required to find out the motive and novelty of the present work by extensively comparing the presented literature in the manuscript. In its current form, The objective of the manuscript is ambiguous. Please re-write it for better comprehension. Also, I suggest writing the objective according to the novelty of the manuscript. Is it a research or state-of the art review paper?
Please, at the end of the introduction, including the structure of the manuscript.
I recommend giving an introduction between sections 2 and 2.1. The same in sections 4 and 4v.1.
Vectorize the figures to see the details, for example, figure 13.
Include quantitative values in the abstract to highlight the findings.
Table 1: Please add headers to the table.
Include quantitative results in the last section.
Section 3: Describe the validation method in this section.
Can you discus in the finite element section other similar works in order to see the differences of the employed software as: Finite-element simulation for thermal modeling of a cell in an adiabatic calorimeter; Finite element method and cut bar method-based comparison under 150ffi, 175ffi and 310 ffiC for an aluminium bar
I suggest adding the mechanical properties of carbon fiber prepreg and epoxy 118 MTM44/CF5804A-40% RW-DC in the appendix section or a table.
Author Response
The manuscript is found with some grammatical and typographical errors. The authors are suggested to go through the manuscript thoroughly and proofread for grammatical and typographical errors.
Reply from Authors: Thank you for suggestion. The manuscript has been revised for grammatical and typographical errors by using the services provided by Paperpal Preflight.
Lines 23-28: References are missing.
Reply from Authors: Thank you for suggestion. References was added to support these sentences in the manuscript.
The objective of the manuscript is ambiguous. Please rewrite it for better comprehension. Also, I suggest writing the objective according to the novelty of the manuscript. Is it a research or state-of the art review paper?
Reply from Authors: Thank you for suggestion. This manuscript is a research paper. The objective aims have been strengthen (as suggested) in the last paragraph of the introduction section of the paper. The objective also describes the novelty based on the presented literature review. As such, in this current study the authors are continuing the work of Pavan by investigating thin-walled top-hat structures made from carbon composites. We are also extending the work of Pavan from door intrusion beams to add on the B-pillars as well. Therefore, this study is just an initial study of top-hat composite structures for transverse loading applications.
Pavan K, S. Nagarajan, Enhancement of car door optimization for crash analysis, Int. J. Pure Appl. Math. 2018, 119–7, 1039–1045
Please, at the end of the introduction, including the structure of the manuscript.
Reply from Authors: Thank you for suggestion. This suggestion was included at the end of the introduction section.
I recommend giving an introduction between sections 2 and 2.1. The same in sections 4 and 4v.1.
Reply from Authors: Thank you for suggestion. This suggestion was included at the relevant sections.
Include quantitative values in the abstract to highlight the findings.
Reply from Authors: Thank you for suggestion. This suggestion was included in the abstract
Table 1: Please add headers to the table.
Reply from Authors: Thank you for suggestion. This suggestion was included in Table 1
Section 3: Describe the validation method in this section.
Reply from Authors: Thank you for suggestion. The authors believe that the validation should remain in Section 4 because it has useful discussions that will support the reason behind using four-point bending.
Can you discus in the finite element section other similar works in order to see the differences of the employed software as: Finite-element simulation for thermal modeling of a cell in an adiabatic calorimeter; Finite element method and cut bar method-based comparison under 150ffi, 175ffi and 310 ffiC for an aluminium bar.
Reply from Authors: Thank you for suggestion. However, the authors are not sure if this comment will enhance the manuscript. In the view of crashworthiness and impact analysis, researchers and designers from industry and academia are well known to either utilize these validated commercial softwares such as ABAQUS, LS-Dyna, Pam-Crash to name a few.
I suggest adding the mechanical properties of carbon fiber prepreg and epoxy 118 MTM44/CF5804A-40% RW-DC in the appendix section or a table.
Reply from Authors: Thank you for suggestion. This is already shown in Table 1
Round 2
Reviewer 2 Report
You need to explain cohesive zone model used in FE-simulation in detail.
Author Response
Reviewer: You need to explain cohesive zone model used in FE-simulation in detail.
Authors: Thank you for the suggestion. This was added to the manuscript. This change is highlighted in green text in the manuscript.